# CIRCUMVENTING NEGATIVE TRANSFER VIA CROSS GENERATIVE INITIALISATION

**Wenjun Bai, Changqin Quan & Zhi-Wei Luo**
Department of Computional Science
Graduate School of System Informatics
Kobe University
Kobe, JAPAN
`{bwj, qcq, lzw}@cs11.cs.kobe-u.ac.jp`

## ABSTRACT

Negative transfer – a special type of transfer learning – refers to the interference of the previous knowledge with new learning. In this research, through an empirical study, we demonstrate the futile defence to the negative transfer via conventional neural network based transfer techniques, i.e., mid-level feature extraction and knowledge distillation. Under a finer specification of transfer learning, we speculate the real culprits of negative transfer are the incongruence on task and model complexity and the ordering of learning. Based on this speculation, we propose a tentative transfer learning technique, i.e., **cross generative initialisation**, to sidestep the negative transfer. The effectiveness of **cross generative initialisation** was evaluated empirically.

## 1 UNAVAILING KNOWLEDGE TRANSFER IN NEGATIVE TRANSFER

In inductive transfer learning (Pan & Yang, 2010), learning a target task is benefited by the transferred knowledge from a previous learned task, i.e., a source task. In neural network based transfer learning, the extracted mid-level features are served as the knowledge to transfer. The production of these transferable features is accomplished by running forward propagation of a trained neural network (Oquab et al., 2014). To optimise the usage of extracted features in transfer learning, Hinton et al (Hinton et al., 2015) proposed a knowledge distillation technique to allow the compression of a cumbersome model to a compact one. The distilled knowledge, i.e., the cross entropy loss of a learned neural network, is augmented to fine-tune a to-be-learned neural network.

However, both techniques are futile in a special case of transfer learning: negative transfer. Negative transfer – a term borrowed from cognitive science – occurs in a situation where the prior learning of a source task interferes with the later learning of a target task (Pan & Yang, 2010). To formalise our discussion on negative transfer, considering two sequentially to-be-learned tasks with varying degrees of complexity; e.g., $T_1$ (a complex task) and $T_2$ (a simple task), we then assign two corresponding models, e.g., $M_1$ (a cumbersome model) and $M_2$ (a compact model) to learn the foregoing tasks. Dependent upon the congruence on task and model complexity and the learning sequence, there are four distinctive transfer learning cases, i.e., $T_1 D_1 \rightarrow T_2 M_2$; $T_1 M_2 \rightarrow T_2 M_1$; $T_2 M_1 \rightarrow T_1 M_2$; $T_2 M_2 \rightarrow T_1 M_1$. [1]

As demonstrated in Table 1, negative transfer can only be circumvented in which the congruence on model and task complexity is high, and the ordering of learning is followed as complex to simple, i.e., $T_1 M_1 \rightarrow T_2 M_2$. In other three cases, both mid-level feature extraction and knowledge distillation techniques failed to sidestep the negative transfer.

---

[1] $\rightarrow$ denotes the direction of knowledge transfer, e.g., $T_1 D_1 \rightarrow T_2 M_2$ means the knowledge is extracted from a prior learning of a complex task through a cumbersome model, then transferred to assist the learning of a simple task through a compact model.

Table 1: Failed transfer learning techniques in negative transfer

| Transfer From.
Transfer To. | T1M1
T2M2 | T2M1
T1M2 | T1M2
T2M1 | T2M2
T1M1 |
|---|---|---|---|---|
| Transfer Means | Testing Accuracy (%) | | | |
| **Mid-level Feature Extraction** | 94.00±0.56 | 68.15±1.10 | 95.76±0.14 | 33.60±3.31 |
| **Knowledge Distillation** | **98.25±0.09** | **95.13±0.07** | 98.19±0.01 | 94.97±0.23 |
| **Without Transfer Benchmark** | 97.03±0.02 | 95.06±0.01 | 98.68±0.02 | 97.65±0.03 |

In *Experiment 1*, $T_1$: a multiclass classification task (MNIST 0-7); $T_2$: a binary classification task (MNIST 8-9); $M_1$: a relative deep neural network (five hidden layers); $M_2$: a relative shallow neural network (one hidden layer). We ran each technique 10 times on above-mentioned four cases. The negative transfer is defined as the attenuated performance on the transferred target task compare to the benchmark, i.e., the learning without transfer.

## 2 CROSS GENERATIVE INITIALISATION TO SIDESTEP NEGATIVE TRANSFER

### 2.1 CROSS GENERATIVE INITIALISATION

Different to early researches on negative transfer, which focused on the task relatedness (Lee et al., 2016) (Mahmud & Ray, 2008), based on our observations in *Experiment 1*, we propose a Bayesian neural network (BNN) based transfer learning technique, i.e., **cross generative initialisation**, to allow the transferred knowledge to be tuned with the congruence between task and model complexity.

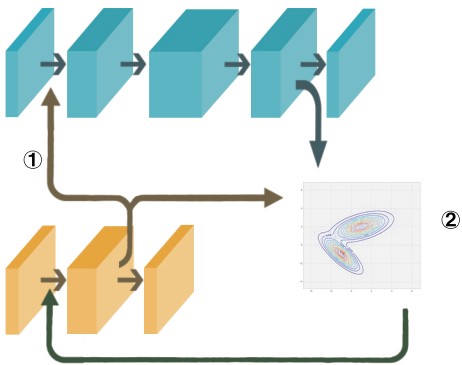

Figure 1: Thumbnail sketch of cross generative initialisation.

In Figure 1, two BNNs, e.g., a compact and a cumbersome BNN, are distinguished by different colours (amber and cyan). For the illustrative purposes, this sketch only depicts the case in $T_1 D_1 \rightarrow T_2 M_2$. The employed GMM in the *generative initialisation* stage (cf. ②) receives two inputs. One is the copy of bootstrapped weights from a target BNN, the other is the posterior weights from a trained source BNN.

In a nutshell, the proposed **cross generative initialisation** is a two-stage initialisation process, i.e., *cross* and *generative* initialisation. As demonstrated in Figure 1, the initial *cross initialisation* (cf. ① in Figure 1) ensures a source BNN is initialised by the non-parametric bootstrapping (Rubin, 1981) on a target BNN. In the sequential *generative initialisation* stage (cf. ② in Figure 1), it treats the variational inferred (Wainwright et al., 2008) posterior weights of a source BNN as an input to a generative model, e.g., a Gaussian mixture model (GMM). The employed GMM is then trained to generate the final transferrable weights in fine-tuning the target BNN. In practice, two GMMs are used separately in generating different transferrable weights for middle and classification layers of the target BNN. The pseudo-code for **cross generative initialisation** is delineated in Algorithm 1.

---

**Algorithm 1** Cross Generative Initialisation

---

**INPUT:** $D_s$: the training dataset for the source task, $M$: the number of bootstrapping points
**OUTPUT:** $\theta_t$: the parameters in the target model

1: **procedure** CROSS GENERATIVE INIT($D_s$; $M$)
2:   **function** *Generative Init*($D_s$)
3:     **for** $d_s \in D_s$ **do**
4:       **Initialise:** $\theta_{init_s}$ : initialise the parameters in the source model
5:       **function** *Cross Init*($M$)
6:         **for** $m \in M$ **do**
7:           $\theta_{bb_t}$: non-parametric bootstrapped on the target model
8:           $\theta_{init_s} \leftarrow \theta_{bb_t}$
9:         **end for**
10:         **return** $\theta_{init_s}$; $\theta_{bb_t}$
11:       **end function**
12:       $\theta_s \leftarrow (\theta_s | \theta_{init_s}, d_s)$ : inferred parameters in the source model based on training
    samples and bootstrapped parameters in the target model
13:     **end for**
14:     **function** GMM(X)
15:       $X \leftarrow concatenate(\theta_{bb_t}; \theta_s)$
16:       **for** $x \in X$ **do**
17:         $\theta_t \leftarrow GMM(x)$: generate the transferrable parameters in a trained GMM
18:       **end for**
19:       **return** $\theta_t$
20:     **end function**
21:     **return** $\theta_t$
22:   **end function**
23: **end procedure**

---

## 2.2   PRELIMINARY EMPIRICAL VALIDATION

To validate our proposed **cross generative initialisation**, we pitted it against the benchmark, i.e., the without transfer learning case. The experimental set-ups inherited straightforwardly from the *Experiment 1*. From the results depicted in Table 2, our proposed **cross generative initialisation** is proved to circumvent the negative transfer.

Table 2: Cross generative initialisation in negative transfer

| *Transform From.* *Transform To.* | T1M1 T2M2 | T2M1 T1M2 | T1M2 T2M1 | T2M2 T1M1 |
|---|---|---|---|---|
| Transfer Means | Testing Accuracy (%) | | | |
| **Cross Generative Initialisation** | **98.48±0.08** | **94.70±0.20** | **97.52±0.09** | **93.03±0.11** |
| **Without Transfer Benchmark** | 97.68± 0.02 | 94.69± 0.01 | 97.32±0.01 | 88.97± 0.02 |

In *Experiment 2*, $T_1$: a multiclass classification task (MNIST 0-7); $T_2$: a binary classification task (MNIST 8-9); $M_1$: a five layer Bayesian neural network with normal prior; $M_2$: a three layer Bayesian neural network with normal prior

## 3   CONCLUSIONS

As a special case of transfer learning, negative transfer has been rarely studied. In this short article, we show the failed conventional transfer learning techniques in defending the negative transfer. Based on the observations in *Experiment 1*, we propose a tentative approach – **cross generative initialisation** – to sidestep the negative transfer. In future research, the theoretical aspects of this proposed **cross generative initialisation** will be fully explored.

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
