# OpenReview forum: "Circumventing negative transfer via cross generative initialisation"
_ICLR.cc/2018/Workshop — Reject_

### Official Review · AnonReviewer1 · 2018-03-09

**Rating:** 4
**Confidence:** 4

**Review:**

Summary: This paper studies the topical problem of transfer learning. Th focus is on proposing a transfer learning method that is more robust to negative transfer (transfer outcome being worse than train from scratch without any transfer), which is a risk in the case of many existing transfer learning methods. The proposed “Cross-Generative” initialisation method uses bootstrapping, and a weight-GMM transferred from the source model. The results show more robust transfer (no negative transfer) on an MNIST classification task using an MLP.

Novelty: Unclear (poorly written/explained, see next)
Clarity: In general, this paper is poorly written and hard to follow, and the proposed methodology is poorly explained. The core methodology is explained in just about 8 lines at the end of Pg 2, which is not enough to assess it. The Algorithm 1 is completely uninformative being mostly composed of control flow statements. Weak English throughout doesn't help.
Significance: The strength of the empirical result is unclear. It’s based on one experiment, and not clear if it generalises. The outcome of this kind of TL experiment depends on all sorts of things like learning rate, etc, so its hard to be convinced that the negative transfer problem shown in Tab 1 is for real, or just the result of bad tuning. Also the ubiquitous pre-train/fine-tune baseline for transfer learning seems not to be compared, which is a fatal gap in a TL paper. (* Unless this is what is meant by “mid-level feature extraction”: but this baseline is not explained, which is another problem).
Quality: Many questionable statements are made. EG:  PG1: “negative transfer can only be circumvented in which the congruence on model and task complexity is high…” this might be the result in Tab 1, but there is definitely not enough evidence/theory to make an absolutely general claim like this. PG 3: “As a special case of transfer learning, negative transfer has been rarely studied”. Negative transfer is more accurately characterised as the failure cases of transfer learning. And it has been studied, many papers try to make TL methods that are more robust to negative transfer.

Assessment: Weak writing, particularly unclearly explained method, and questionable statements. There may be novel ideas in there somewhere, but the current execution is substandard for ICLR Workshop.

---

### Official Review · AnonReviewer2 · 2018-03-11
**not fully clear**

**Rating:** 3
**Confidence:** 5

**Review:**

This paper tackles the problem of negative transfer, a problem that rises every time knowledge transfer
is applied between tasks that are only marginally related among each other. Here the authors show that
standard transfer learning solutions like extracting and re-using cnn-mid-level features or distillation
techniques do not have any safe guard against the negative transfer issue. As an alternative solution, this
work proposes to interconnect the source and target task networks in a new way where their intermediate
parameters are used to train a GMM that then is abe to generate new and transferrable parameters to
fine-tune the target task network without negative transfer.

There are some points in the paper that need more attention and a more detailed explanation

1) From table 1 it can be observed that when M1 is in source knowledge distillation works well regardless of starting
from T1 or T2. This is not surprising since knowledge distillation is designed for a transfer from a complex teacher
network to a simple student network. The performance drops if M2 is in source. On the other side, the mid-level feature
extraction strongly depends on which is the source task since it is designed for passing from a complex large task
to a smaller and simpler one. This is to say that the underlying issue is not just negative transfer related to how much
the observed categories in source and target are similar, but the fact that existing methods are applied in cases out
of their original design conditions.

2) I find also a bit confusing the idea that the source and target network can exchange information among each
other while both are in training. The original setting for transfer learning supposes that the source model is already
trained and the target model is learned only in a second step. If both source and target data are available at the
same time, why not putting the data together to define an overall task or why not attempting a co-training process?
Again it seems that the deigned coditions for transfer learning are not respected.

3) I'm not convinced by the results in table 2. First of all the advantage of the proposed method with respect to the
no-transfer baseline is not significant in the T2M1->T1M2 case and is anyway less than 1% in the T1M1->T2M2 and
T1M2->T2M1 cases. Moreover, it is not clear the comparison with the results presented in table 1: the baseline results
change in the two tables so there is no clear reference that we can use to benchmark and the results of the
competing distillation and feature transfer approaches seems also higher in table 1 for some cases with respect to those
of the proposed method in table 2.

Minor:
In different points D1 is used instead of M1 creating some confusion

---

### Official Review · AnonReviewer3 · 2018-03-14
**An heuristic method to solve the negative transfer problem in CNN training**

**Rating:** 4
**Confidence:** 5

**Review:**

This paper proposed the problem of negative transfer for CNN training, which has not been well studied.
The paper further propose a heuristic method to solve the so-called negative transfer problem.
The problem is interesting, but I don't think the negative transfer problem has been well defined and justified. It seems that the results is not better, then we call it negative transfer. However, in the setting of the deep learning, even for the fine-tune procedure, there are many parameters and methods that can be tried. It is difficult to justify whether the poor performance is due to improper choosing of the algorithms or the problem does have the issue of negative transfer.
What's more, the authors just use the MNIST as examples, which is quite difficult to claim the general negative transfer problem. The proposed approach is also heuristic, and lacks solid theoretical supporting.

---

### Decision · Program_Chairs · 2018-03-20
**ICLR 2018 Workshop Acceptance Decision**

**Decision:**

Reject

**Comment:**

Based on the reviews, this paper has not been accepted for presentation at the ICLR workshop. However, the conversation and updates can continue to appear here on OpenReview.